# Rational Metabolic Engineering Combined with Biosensor-Mediated Adaptive Laboratory Evolution for L-Cysteine Overproduction from Glycerol in *Escherichia coli*

Xiaomei Zhang [1], Zhenhang Sun [1], Jinyu Bian [1], Yujie Gao [2], Dong Zhang [1], Guoqiang Xu [2], Xiaojuan Zhang [2], Hui Li [1], Jinsong Shi [1] and Zhenghong Xu [2,*]

[1] Laboratory of Pharmaceutical Engineering, School of Life Science and Health Engineering, Jiangnan University, Wuxi 214122, China; zhangxiaomei@jiangnan.edu.cn (X.Z.); 6211507033@stu.jiangnan.edu.cn (Z.S.); 6191502001@stu.jiangnan.edu.cn (J.B.); d.zhang1993@outlook.com (D.Z.); lihui@jiangnan.edu.cn (H.L.); shijs@jiangnan.edu.cn (J.S.)

[2] National Engineering Research Center for Cereal Fermentation and Food Biomanufacturing, Jiangnan University, Wuxi 214122, China; 7200201006@stu.jiangnan.edu.cn (Y.G.); xuguoqiang@jiangnan.edu.cn (G.X.); zhangxj@jiangnan.edu.cn (X.Z.)

* Correspondence: zhenghxu@jiangnan.edu.cn; Tel./Fax: +86-510-85918206

**Abstract:** L-Cysteine is an important sulfur-containing amino acid with numerous applications in the pharmaceutical and cosmetic industries. The microbial production of L-cysteine has received substantial attention, and the supply of the precursor L-serine is important in L-cysteine biosynthesis. In this study, to achieve L-cysteine overproduction, we first increased L-serine production by deleting genes involved in the pathway of L-serine degradation to glycine (serine hydroxymethyl transferase, SHMT, encoded by *glyA* genes) in strain 4W (with L-serine titer of 1.1 g/L), thus resulting in strain 4WG with L-serine titer of 2.01 g/L. Second, the serine-biosensor based on the transcriptional regulator NCgl0581 of *C. glutamicum* was constructed in *E. coli,* and the validity and sensitivity of the biosensor were demonstrated in *E. coli.* Then 4WG was further evolved through adaptive laboratory evolution (ALE) combined with serine-biosensor, thus yielding the strain 4WGX with 4.13 g/L L-serine production. Moreover, the whole genome of the evolved strain 4WGX was sequenced, and ten non-synonymous mutations were found in the genome of strain 4WGX compared with strain 4W. Finally, 4WGX was used as the starting strain, and deletion of the L-cysteine desulfhydrases (encoded by *tnaA*), overexpression of serine acetyltransferase (encoded by *cysE*) and the key enzyme of transport pathway (encoded by *ydeD*) were performed in strain 4WGX. The recombinant strain 4WGX-Δ*tnaA*-*cysE*-*ydeD* can produce 313.4 mg/L of L-cysteine using glycerol as the carbon source. This work provides an efficient method for the biosynthesis of value-added commodity products associated with glycerol conversion.

**Keywords:** *Escherichia coli*; biosensor; glycerol; adaptive laboratory evolution; L-cysteine

## 1. Introduction

L-cysteine has been widely used in the food, agricultural and pharmaceutical industries. Because of the toxicity of L-cysteine and the complex regulation of its synthesis pathway, efficient microbial production of L-cysteine at the industrial scale has not been achieved [1–4]. Most studies have focused on producing L-cysteine from glucose by recombinant *Escherichia coli* or *Corynebacterium glutamicum* [5–8]. However, *C. glutamicum* grows slowly, thus resulting in a long manufacturing cycle. Compared with *C. glutamicum*, *E. coli* has a higher growth rate, and the its genetic engineering method is well developed, thus suggesting that the production of L-cysteine by *E. coli* has great potential [2,3,7,8]. The precursor of L-cysteine is L-serine in *E. coli*, and the biosynthesis of L-cysteine from L-serine in *E. coli* occurs via a two-step pathway, the catalysis of L-serine acetyltransferase

(encoded by *cysE*) and L-cysteine synthase (encoded by *cysK*) (Figure 1). The first reaction catalysed by CysE is the rate limiting step of L-cysteine biosynthesis in *E. coli*. Moreover, multiple L-cysteine desulfhydrases (CD encoded by *tnaA*) catalyse the degradation of L-cysteine [2,3,7–9]. Previous studies have shown that L-serine is an important precursor for the biosynthesis of L-cysteine, and enhancing the L-serine synthesis is a necessary metabolic engineering strategy for L-cysteine accumulation [1,2].

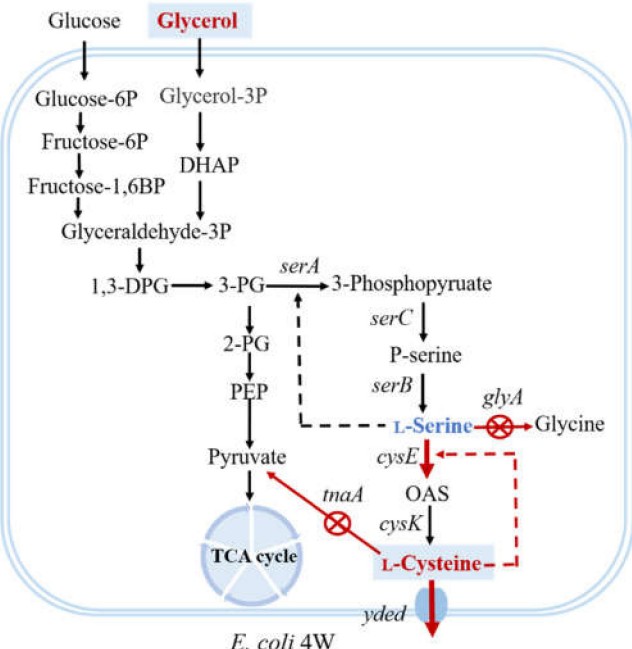

**Figure 1.** The protocol of constructing strain 4WGX over-producing L-cysterine from glycerol. *glyA* encoded serine hydroxymethyl transferase, *tnaA* encoded L-cysterine transporter, Red crosses on solid lines (⊗) indicated genes that were deleted. *cysE* encoded serine acetyltransferase, the red line indicated gene that were overexpressed. The starting strain *E. coli* 4W had been constructed in our previous study with deletion of *sdaA, sdaB,* and *tdcG* (The three genes encoded L-serine deaminases), and the removal of feedback inhibition of *serA* (*serA* encoded 3-phosphoglycerate dehydrogenase).

Simultaneously, in addition to traditional fermentation substrates such as glucose, sucrose and other sugar raw materials, glycerol has become a very competitive new choice [10,11]. As non-renewable fossil energy is increasingly depleted, searching for new alternative energy sources, such as biodiesel, has become a top priority. However, the large-scale development and utilization of biodiesel has brought about another serious problem: the treatment and reuse of crude glycerol, which was the by-product of biodiesel manufacturing. According to statistics, every 10 kg of biodiesel produces approximately 1 kg of crude glycerol by-product [12,13]. If these by-products could be converted into high value-added chemicals, the crude glycerol could be reused, and the chemical cost could also be decreased. Therefore, using glycerol as a carbon source to produce high value-added chemicals has become a major topic in the biodiesel industry [14,15]. The chemicals produced by using glycerol as a substrate mainly include shikimic acid, lactic acid, succinate, lysine, L-phenylalanine and 1,3-propanediol [11,13,16–19]. However, nearly all studies on the production of L-cysteine by *E. coli* have focused on the pathway begin from glucose. Compared with that from glucose, the metabolic pathway from glycerol to L-cysteine is shorter, and the carbon atom economy is also better [20,21].

ALE, also called adaptive evolutionary engineering, has become a valuable tool in metabolic engineering for strain development [22]. However, the traditional screening procedure is cumbersome and time-consuming. A pressing need to develop an efficient approach to screen high-yield mutant strains [23–25]. High-throughput screening methods

with biosensors have been used to screen mutant strains, such as those overproduce amino acids and organic acids, and the biosensors, including the ones based on riboswitches, enzymes, and transcription factors, can transform information about a specific metabolite into a graded fluorescence output [26–31]. In our previous study [32], we developed a genetic metabolite biosensor capable of detecting L-serine in single *C. glutamicum* cells, this biosensor is based on the transcriptional regulator *NCgl0581* of *C. glutamicum*, which activates expression of the *NCgl0580* promoter to drive transcription of enhanced yellow fluorescent protein [32]. However, no study has reported screening of L-serine overproducing strains a biosensor in *E. coli*.

In this study, to achieve L-cysteine overproduction, we first enhanced production of the precursor L-serine by deleting the L-serine degradation pathway *glyA* with CRISPR/Cas9 in strain 4W, thus resulting in strain 4WG. Second, in *E. coli*, we constructed a serine-biosensor based on the transcriptional regulator NCgl0581 of *C. glutamicum*. The validity and sensitivity of the biosensor were studied. Subsequently, 4WG was further evolved by using ALE combined with serine-biosensor, thus yielding the evolved strain 4WGX. The whole genome of the evolved strain 4WGX was sequenced, and comparative genomics analysis and reverse mutation were performed. Finally, 4WGX was used as the starting strain, and the deletion of the L-cysteine desulfhydrases (*tnaA*), overexpression of serine acetyltransferase (*cysE*) and the key enzyme in the transport pathway (*ydeD*) were performed. The recombinant strain was successfully constructed and found to produce L-cysteine using glycerol as substrate.

## 2. Materials and Methods

### 2.1. Strains and Plasmids

Strains and plasmids used in this study are summarized in Table 1. Primers for gene cloning and deleting are listed in Table 2. Strain 4W, carrying deletions of *sdaA*, *sdaB* and *tdcG*, was constructed in our previous study [20]. The serine-biosensor pDser from *C. glutamicum* was also constructed in our previous study [32]. The plasmids pTarget and pCas were used for knocking out the *glyA* gene.

**Table 1.** Strains and plasmids used in this study.

| Strains or Plasmids | Description | Sources |
|---|---|---|
| **Strains** | | |
| *E. coli* JM109 | recA1, endA1, gyrA96, thi-1, hsd R17(rk- mk+) supE44 | Invitrogen |
| 4W | W3110△*tdcG*△*sdaA*△*sdaB serA*^dr | Invitrogen |
| 4WG | 4W with *glyA* deletion | This study |
| 4W-pDer | 4W harboring serine-biosensor pDser | This study |
| 4WGX | A mutant derived from 4W | This study |
| 4WG-pDer | 4WG harboring serine- biosensor pDser | This study |
| 4WG-*cysE* | 4WG harboring pEtac-*cysE* | This study |
| 4WG-*cysE-ydeD* | 4WG harboring pEtac-*cysE-ydeD* | This study |
| 4WG-Δ*tnaA* | 4WG with *tnaA* deletion | This study |
| 4WG-Δ*tnaA-cysE* | 4WG-Δ*tnaA* harboring pEtac-*cysE* | This study |
| 4WG-Δ*tnaA-cysE-ydeD* | 4WG-Δ*tnaA* harboring pEtac-*cyE-ydeD* | This study |
| **Plasmids** | | |
| pCas | Carrying Cas9 and λRed System, kan | Invitrogen |
| pTargetF | Carrying N20 sequence, spc or smr | Invitrogen |
| pDser | Biosensor, kan | Invitrogen |
| pEtac | Inducible expression plasmid, *tac*, kan | This study |
| pEtac-*cysE* | Carrying *cysE* gene from *E. coli* | This study |
| pEtac-*cysE-ydeD* | Carrying *cysE* and *ydeD* gene from *E. coli* | This study |

**Table 2.** Primers used in this study.

| Primers | Sequence |
| --- | --- |
| pTargetF-△*glyA*1-F | ACTGTGGCAGGCTATGGAGCGTTTTAGAGCTAGAAATAGCAAGTT |
| pTargetF-△*glyA*1-R | GCTCCATAGCCTGCCACAGTACTAGTATTATACCTAGGACTGAGC |
| pTargetF-△*glyA*2-F | AGAAGCCGAAGCGAAAGAACGTTTTAGAGCTAGAAA-TAGCAAGTT |
| pTargetF-△*glyA*2-R | GTTCTTTCGCTTCGGCTTCTACTAGTATTATACCTAGGACTGAGC |
| *glyA*-U-F | AGCCCTGCAATGTAAATGGTT |
| *glyA*-U-R | ACAGCAAATCACCGTTTCGCCCGCATCTCCTGACTCAGCTA |
| *glyA*-D-F | AGCTGAGTCAGGAGATGCGGGCGAAACGGTGATTTGCTGTC |
| *glyA*-D-R | TCGCCAGACAGGATTTAACCC |
| pTargetF:1756F23 | CCCTGATTCTGTGGATAACCGTA |
| pTargetF:78R23 | ACATCAGTCGATCATAGCACGAT |
| *cysE*-F | TTCACACAGGAAACAGAATTCATGTCGTGTGAAGAACTGGAAATTG |
| *cysE*-R | TGCGGCCGCAAGCTTGTCGACTTAGATCCCATCCCCATACTCAA |
| *ydeD*-F | GGGATCTAAGTCGACAAGCTTCGCTGAGCAATAACTAGCATAACC |
| *ydeD*-R | GTGGTGGTGGTGGTGCTCGAGTTAACTTCCCACCTTTACCGCT |
| *tnaA*-U-F | TTGCATATATATCTGGCGAATTAATCGG |
| *tnaA*-U-R | GCCACTCTGTAGTATTAAGTATCAAAGAAATAGTTAGAGAACGCCA |
| *tnaA*-D-F | ACTTAATACTACAGAGTGGCTATAAGGATGTT |
| *tnaA*-D-R | ACGAAAATGGCTGTGCAGAT |
| pTargetF-Δ*tnaA*-F | CGTTCTCTTTCACATGTTAACTAGTATTATACCTAGGACTG |
| pTargetF-Δ*tnaA*-R | TAAACATGTGAAAGAGAACGTTTTAGAGCTAGAAATAGCAA |

## 2.2. Growth Medium and Culture Conditions

Luria-Bertani (LB) medium was used for plasmid construction. When appropriate, streptomycin (50 μg/mL) or kanamycin (50 μg/mL) was added. For ʟ-serine fermentation, mineral AM1 medium [33] supplemented with 10.0 g/L glycerol, 8.6 g/L $(NH_4)_2 \cdot HPO_4$, 3.9 g/L $NH_4H_2PO_4$ and 1 g/L yeast extract was used. *E. coli* was cultured according to our previous study [20].

## 2.3. Gene Deletion with CRISPR/Cas9

For deletion of *glyA* gene, as an example, the upstream homologous arms of the *glyA* gene were obtained by using primers *glyA*-U-F and *glyA*-U-R, and the downstream homologous arms of *glyA* gene by using primers *glyA*-D-F and *glyA*-D-R. Then both the upstream and downstream arms were used as templates, and the homologous recombination repair templates were obtained by overlap extension PCR using primers *glyA*-U-F and *glyA*-D-R.

Plasmid pTarget was extracted from *E. coli* JM109 and amplified with primers pTargetF-△*glyA*1-F, pTargetF-△*glyA*1-R and pTargetF-△*glyA*2-R and pTargetF-△*glyA*2-F. The template was degraded with the *Dpn*I enzyme, and then PCR products were transferred to *E. coli* JM109 to repair the cyclization gap. After the clones were grown, pTargetF:1756F23 and pTargetF:78R23 were used for PCR amplification. If the sequencing results were correct, then plasmids were amplified and extracted. Finally, two pTarget plasmids containing N20 sequences specifically targeting *glyA* were obtained. The N20 sequences were predicted in CHOPCHOP (chopchop.cbu.uib.no).

Finally, homologous arm fragments and two pTarget plasmids were electroporated into competent cells containing the pCas plasmid. Transformants were selected on kanamycin and streptomycin plates and verified by PCR using the corresponding primers *glyA*-U-F and *glyA*-D-R. The gene *tnaA* was deleted by this method.

## 2.4. Gene Overexpression

The plasmid pEtac was used for the expression of foreign genes. This plasmid carries a tac promoter and Kan resistance marker. For the expression of *cysE*, as an example, we amplified the *cysE* gene by using the primers c*ysE*-F/*cys*-R. The target gene was digested with *EcoR* I and *Sal* I, then ligated with the linearized plasmid pEtac to construct the

inducible expression plasmid pEtac-*cysE*. The gene *ydeD* was overexpressed through this method, thus yielding the plasmid pEtac-*cysE-ydeD*.

### 2.5. Construction and Verification of the Serine-Biosensor

The pDser plasmid was constructed in our previous study [32]. The pDser plasmid was introduce into *E. coli* 4WG, transformants were selected and verified, and strains containing the pDser plasmid were achieved. The verification of the serine-biosensor was performed according to our previous study [32].

### 2.6. Biosensor-Driven Evolution Experiment

For strain evolution, *E. coli* 4WG harboring pDser was cultured in LB medium, and 10% (v/v) inoculum was transferred into fresh AM1 medium with 6 g/L L-serine for 24 h. This step was repeated ten times (recorded as ten generations). The evolved strain was approximately 600 generations. Then L-serine concentration were increased to 12, 25 and 50 g/L, and this process was repeated. Successive rounds of ALE were carried out with the L-serine increased stepwise (6, 12, 25 and 50 g/L). According to the generation time (GT) of *E. coli*, the evolved strain was approximately 600, 1200, 1800, 2400 generations, respectively.

At the end of the experiment, an appropriate amount of the evolved strains after 600, 1200, 1800, 2400 generations were diluted to $10^{-5}$-$10^{-6}$ fold. Then 100 μL of diluted bacterial solution was spread on kanamycin plates and cultured overnight at 37 °C and single colonies were transferred to 96-well plates. After 24 h of fermentation, the fluorescence intensity of the strain in each well was measured, the most efficient strain was selected by using FACS according to our previous study [32].

### 2.7. Genome Sequencing

The whole genome of *E. coli* 4WGX was sequenced, and comparative genomics analysis was performed with the parent strain 4WG. Genomic DNAs of the strains were extracted using Molpure Bacterial DNA Kit (Yeasen. Shanghai, China). Library construction and genome sequencing were performed by Genewiz (Suzhou, China) by using Illumina Hiseq2500 sequencing platform. Quality assurance of the output was analyzed by using FastQC software (v.0.10.1) and NGSQC Toolkit software (v.2.3.3). BWA alignment software (v.0.7.17) and SAM tools software (v.1.9) were used for alignment and variant calling, respectively. Variations were annotated by using the SnpEff software (v.4.3i).

### 2.8. Analytical Methods

Cell growth was measured as the $OD_{600}$ (AOE UV-1200S, China). A triglyceride assay kit for measuring glycerol concentration was purchased from Nanjing Jiancheng Bioengineering Institute. First, a standard curve based on different concentrations of glycerol standard and the corresponding $OD_{550}$ values was constructed, and then the actual glycerol concentration of each sample was calculated according to the $OD_{550}$ value. The fluorescence intensity of bacteria was detected with a microplate reader with an excitation wavelength of 488 nm and emission wavelength of 530 nm. The concentrations of L-serine and L-cysteine were determined with high-performance liquid chromatography (HPLC; Agilent 1100, USA) according to a previously reported method [32].

## 3. Results

### 3.1. Improved the Precursor L-Serine Accumulation by Decreasing L-Serine Degradation in E. coli

To achieve L-cysteine overproduction, we first enhanced the L-serine production of strain 4W. In the L-serine degradation pathway, SHMT (*glyA*) were deleted in strain 4W by using CRISPR/Cas9, thus resulting in strain 4WG. As shown in Figure 2, strain 4WG showed cell growth inhibition, with a maximum $OD_{600}$ of 2.37 (Figure 2b). In contrast, the maximum $OD_{600}$ of parent strain 4W was 5.73 (Figure 2a), *glyA* deletion significantly decreased the cell growth. Correspondingly, the L-serine accumulation of strain 4WG was 0.75 g/L, a level significantly lower than that of the parental strain 4W (1.1 g/L).

We inferred that intracellular glycine deficiency caused by knocking out the pathway of L-serine degradation resulted in poorer growth status of the strain, and leaded to lower L-serine accumulation.

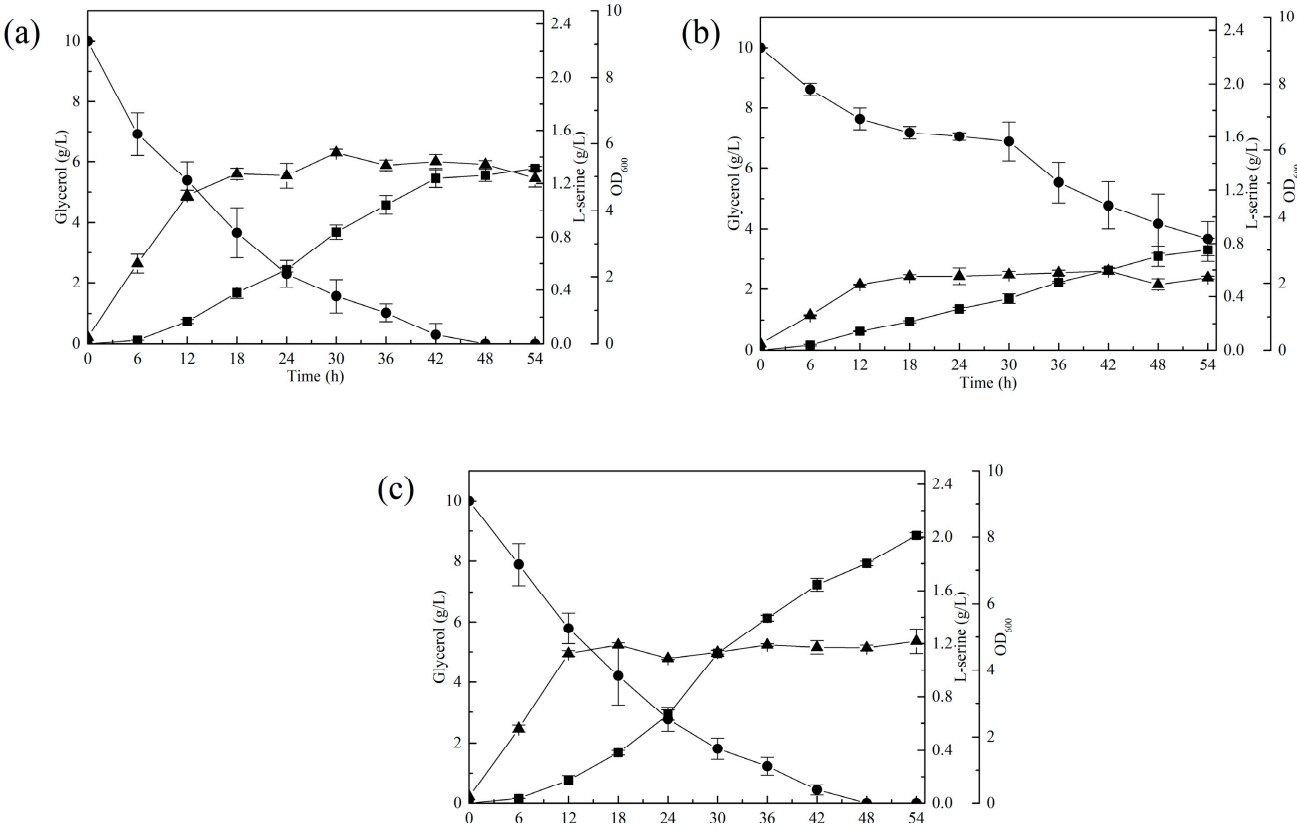

**Figure 2.** Fermentation profiles of strain 4W, strain 4WG and strain 4WG with 0.15 g/L glycine. (**a**) Profiles of glycerol consumption, cell growth and L-serine production in strain 4W; (**b**) Profiles of glycerol consumption, cell growth and L-serine production in 4WG; (**c**) Profiles of glycerol consumption, cell growth and L-serine production in strain 4WG with 0.15 g/L glycine added. Squares represent cell growth, circles represent residual glycerol, and triangles represent L-serine. Values denote the average of three independent experiments, and error bars indicate standard deviation.

According to a previous study [5], inactivation of SHMT in *E. coli* could effectively reduce the intracellular degradation of L-serine, and exogenous glycine could be added to maintain the cell growth. After 0.15 g/L (2 mM) glycine was added to the medium, the strain 4WG returned to normal growth with a maximum $OD_{600}$ value of 4.87 (Figure 2c). Meanwhile, L-serine accumulation also increased significantly, reaching 2.01 g/L after 54 h of fermentation, which was 53.4% higher than that of the control strain 4W. Simultaneously, the substrate glycerol was completely consumed during fermentation for 48 h, which was similar to that for the parental strain 4W.

Although L-serine titer of strain 4WG increased with the addition of glycine, L-serine was found to be highly toxic to this strain even at low concentrations. As shown in Figure 3, the cell growth of strain 4WG was significantly decreased with the addition of L-serine in the medium. When 6 g/L L-serine was added, the maximum $OD_{600}$ was 2.62. When L-serine addition reached 12, 25 and 50 g/L, strain 4WG showed negligible growth. The strain's tolerance to L-serine was the key to over-producing L-serine.

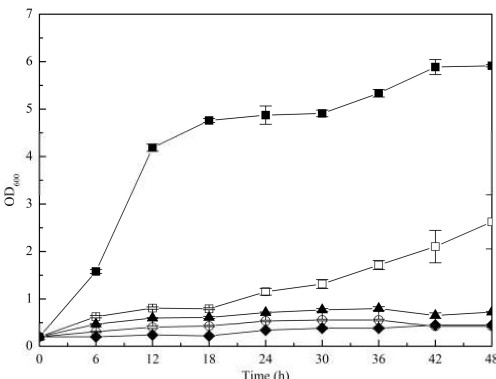

**Figure 3.** Growth profiles of strains 4WG in AM1 containing different concentrations of L-serine. Squares represent 0 g/L L-serine, open squares represent 6 g/L L-serine, triangles represent 12 g/L L-serine, open circles represent 25 g/L L-serine, and diamonds represent 50 g/L L-serine. Values denote the average of three independent experiments, and error bars indicate standard deviation.

### *3.2. Increased L-Serine Production through ALE Combined with a Serine-Biosensor*

### 3.2.1. Construction and Verification of a Serine-Biosensor in *E. coli*

ALE was selected to improve L-serine tolerance and L-serine production. We constructed a serine-biosensor to increase the screening efficiency, and verified its efficacy in *E. coli*. The serine-biosensor of *C. glutamicum* was constructed and used to screen L-serine overproducing strains in our previous study [32]. However, there is no research adapting this biosensor in *E. coli*, and whether the heterologous expression of the biosensor was also effective in high-performance screening in *E. coli* needed to be tested. The serine-biosensor pDser was transformed into *E. coli* 4WG, resulting in 4WG-pDser. Afterward, 4WG and 4WG-pDser were photographed under a laser scanning confocal microscope under visible light and UV light. As shown in Figure 4, the parental strain 4WG showed no fluorescence signal, and 4WG-pDser showed substantial fluorescence intensity. This result confirmed that the serine-biosensor was successfully expressed in *E. coli*.

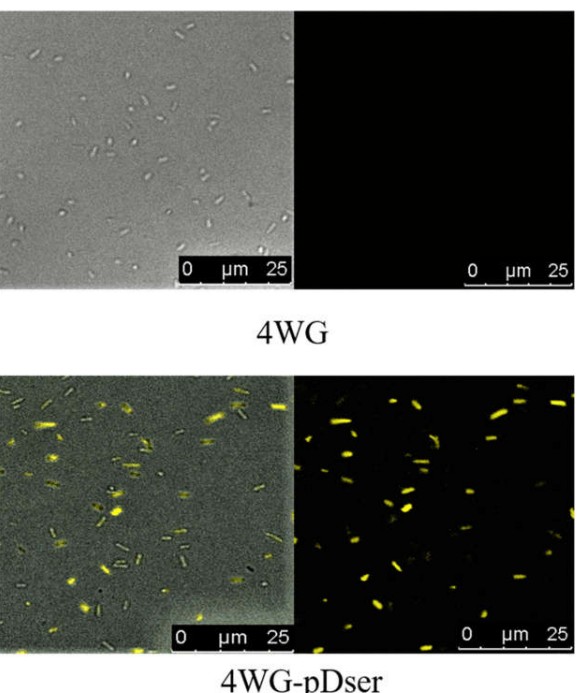

**Figure 4.** Identification of serine-biosensor pDser in *E. coli*. 4WG-pDser clearly emitted yellow fluorescence, whereas the control *E. coli* 4WG did not emit yellow fluorescence.

The relationship between fluorescence intensity and L-serine titer was then studied. The fluorescence signal from the serine-biosensor correlated with the L-serine titer (Figure 5a). Moreover, in the ALE experiment, L-serine was added to the medium, and the effect of the L-serine addition to the biosensor was studied. No fluorescence significant change was observed with varying amounts of L-serine added (data not shown), indicating that only the cellular L-serine biosynthesized was monitored by serine-biosensor. These results demonstrated the functionality of the serine-biosensor in *E. coli*, which was adapted from *C. glutamicum*. We used this method to screen serine over-producing strain in the rest of this study.

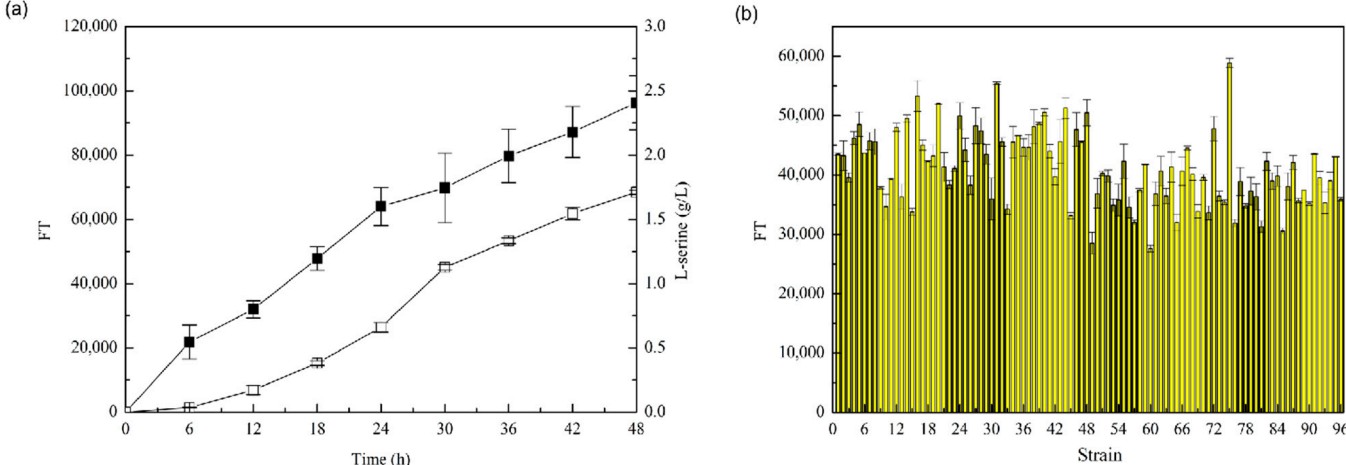

**Figure 5.** The relationship of L-serine accumulation and the fluorescence intensity. (**a**) Comparison of L-serine accumulation and the fluorescence intensity of *E. coli* 4WG-pDser. Squares represent fluorescence intensity, and open squares represent L-serine titer by HPLC; (**b**) Comparison of the adaptive strains' fluorescence intensity. Cylinders represent fluorescence intensity. Values denote the average of three independent experiments, and error bars indicate standard deviation.

3.2.2. Increased L-Serine Yield Achieved by Biosensor-Driven Evolution

ALE with biosensors was used to improve L-serine tolerance and L-serine production. The strain harbouring serine-biosensor (4WG-pDser) was evolved in the medium with 6, 12, 25 and 50 g/L L-serine, and the evolve strain was approximately 600, 1200, 1800 and 2400 generations respectively, according to the generation time of *E. coli*. In this process, we observed that the cell growth of strain 4WG was significantly inhibited in the medium with 25 g/L and 50 g/L L-serine. With increasing generation number, the cell growth rate clearly increased, as did the maximum $OD_{600}$ increased (data not shown). The final evolved strain was achieved 2400 generations. As shown in Figure 5b, the first strain was the control strain 4WGX-pDser, and the remaining 95 strains were single colonies selected on the plates. Five strains with the highest intensity values were selected for flask fermentation. The resultant ALE strain was named 4WGX. As shown in Figure 6a, after 48 h of fermentation, the maximum $OD_{600}$ of strain 4WGX was 6.87, glycerol was completely consumed at 24 h, the L-serine titer was 4.13 g/L at 48 h, which was 105% higher than that of 4WG (2.01 g/L) and 275% higher than that of 4W (1.1 g/L), and the substrate conversion rate was 41.3%. Strain 4WGX was cultured in medium with the addition of 50 g/L L-serine, the cell growth was shown in Figure 6b, the maximum $OD_{600}$ of strain 4WGX reached 3.65, and the parental strain 4WG showed almost no growth in the same medium (Figure 3).

To clarify the reasons for the greatly improved serine-tolerance of strain 4WGX, we sequenced the whole genome of strain 4WGX. The sequencing results revealed a total of eleven single base mutations in the genome of strain 4WGX compared with strain 4W, including ten non-synonymous mutations (*bamA, brnQ, ybcJ, fepB, agp, dgcT, oppB, fliK, ygbN and eno*) and one synonymous mutation (*fdrA*). We chose two genes (*agp* encoded glucose-1-phosphatase, and *eno* encoded enolase) not involved in the membrane for further study. First, reverse mutation of *agp* and *eno* was performed in the genome of 4WG, thus

yielding mutant strains. However, we did not observe any significant change in L-serine production, cell growth and glycerol consumption in both strains (data not shown).

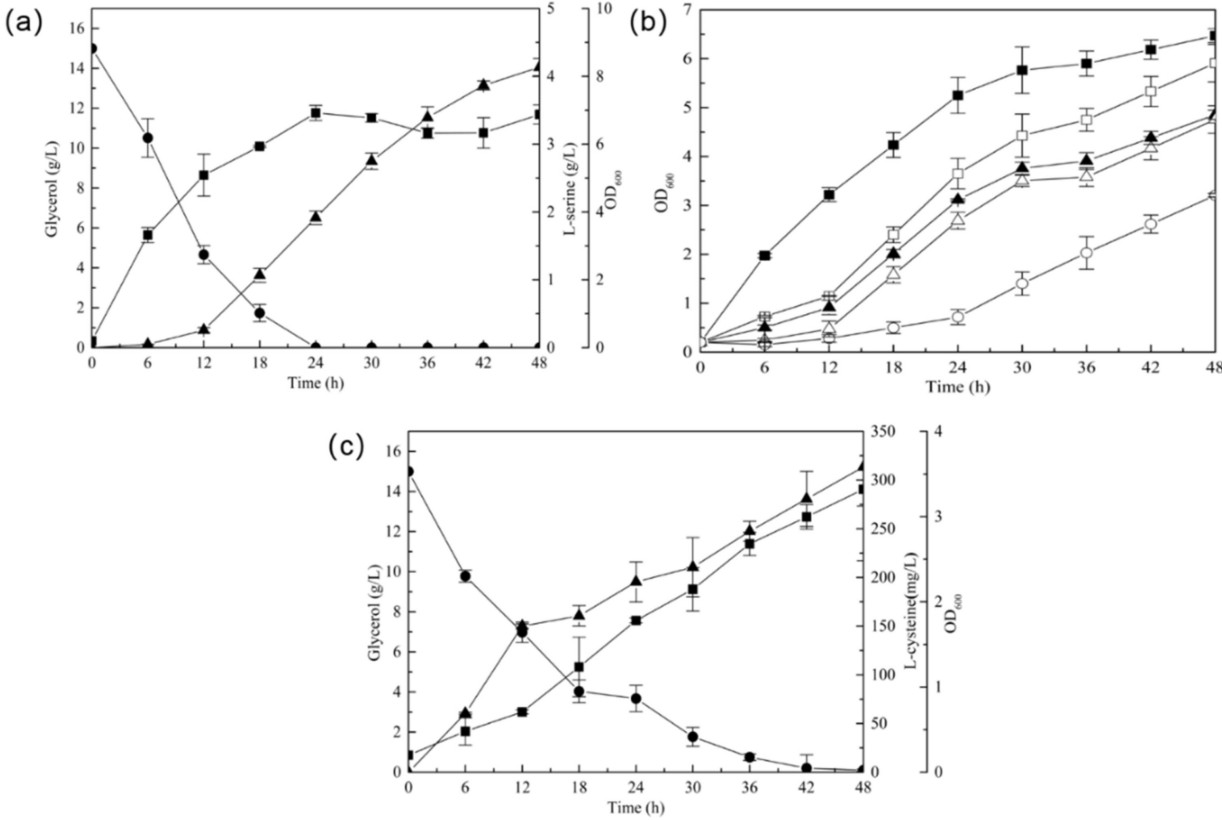

**Figure 6.** Fermentation profiles of strain 4WGX, strain 4WGX in AM1 containing different concentrations of L-serine, and stain 4WGX-Δ*tnaA-cysE-ydeD*. (**a**) Profiles of glycerol consumption, cell growth and L-serine production in strain 4WGX. Squares represent cell growth, circles represent residual glycerol, and triangles represent L-serine; (**b**) Growth profiles of the evolved strains 4WGX in AM1 containing different concentrations of L-serine. Squares represent 0 g/L L-serine, open squares represent 6 g/L L-serine, triangles represent 12 g/L L-serine, open triangles represent 25 g/L L-serine, and open circles represent 50 g/L L-serine; (**c**) Profiles of glycerol consumption, cell growth and L-serine production in strain 4WGX-Δ*tnaA-cysE-ydeD*. Squares represent cell growth, circles represent residual glycerol, and triangles represent L-cysteine. Values denote the average of three independent experiments, and error bars indicate standard deviation.

*3.3. Construction of L-Cysteine-Producing Recombinant Strain*

Strain 4WGX was used as the staring strain, in which the L-cysteine desulfhydrases (*tnaA* gene) gene was deleted by the CRISPR/Cas9 method. The 4WGX genome was used as a template, and primers *tnaA*-1/*tnaA*-2 and *tnaA*-3/*tnaA*-4 were used to amplify the upstream and downstream fragments. The pTarget plasmid with the specific N20 sequence and the homology arm fragment were electroporated into competent cells containing the Cas9 plasmid. Clones were selected on LB agar plates containing kanamycin and streptomycin (50 μg/mL). The clones were verified by PCR, the plasmid was finally eliminated, and strain 4WGX-△*tnaA* was achieved. Then the plasmid pEtac was used for the overexpression of *cysE* and *ydeD*. The plasmid pEtac-*cysE-ydeD* was constructed and transformed into 4WGX-△*tnaA*, thus yielding the L-cysteine-producing strain 4WGX-△*tnaA*-pEtac-*cysE-ydeD*. Fermentation by strain 4WGX-△*tnaA*-pEtac-*cysE-ydeD* was performed (Figure 6c). The cell growth (OD$_{600}$) reached a maximum value of 3.4 at 48 h, the glycerol was completely consumed at 42 h, and the accumulation of L-cysteine gradually increased after 6 h, exhibiting a cell growth-independent production profile. At 48 h, the final L-cysteine titer was 313.4 mg/L, and the original strain 4WGX did not produce L-cysteine. Compared

with those of the original strain 4WGX (Figure 6a), the cell growth and the glycerol consumption rate of 4WGX-△*tnaA*-pEtac-*cysE-ydeD* were significantly lower, thus suggesting that L-cysteine might be toxic to the cell, and improving the strain's tolerance to L-cysteine might be key in the future.

## 4. Discussion

In this study, we successfully established a biosensor-driven laboratory evolution approach using serine-biosensor from *C. glutamicum* for improving L-cysteine production in *E. coli*. Within several iterative rounds, *E. coli* 4WGX was isolated from a large evolved strain library and found to produce 4.13 g/L L-serine. Furthermore, the L-cysteine producing strain was obtained through deletion of *tnaA*, overexpression of *cysE* and *ydeD* in strain 4WGX. The recombinant strain 4WGX-Δ*tnaA*-*cysE-ydeD* with 313.4 mg/L of L-cysteine was constructed using glycerol as the carbon source. This is the first report of producing L-cysteine from glycerol. Compared with glucose as carbon source for microbial L-cysteine production, using glycerol has several advantages, including its better carbon atomic economy and higher degree of reduction; glycerol is a highly promising substrate for amino acid production [20]. Although these results showed that L-cysteine titer was lower than that with glucose as the carbon source, glycerol is an alternative substrate providing a variety of economic and metabolic advantages. With further engineering and optimization, fermentation directly using glycerol as carbon source could become competitive. Moreover, this work provides an efficient method for value-added products bioconversion using glycerol as substrate.

L-serine is the precursor of L-cysteine in *E. coli*, and L-serine accumulation is important to efficient produce L-cysteine. However, degradation is a crucial issue in microbial L-serine production. L-serine has two main degradation pathways to either glycine or pyruvate. The conversion of L-serine to pyruvate in *E. coli* is catalyzed by three L-serine deaminases, *sdaA*, *sdaB* and *tdcG*. The conversion of serine to glycine is catalyzed by serine hydroxymethyl transferase (SHMT). Strain 4W was obtained by deleting *sdaA*, *sdaB* and *tdcG* in *E. coli* W3110. In this study, to remove the L-serine degradation pathway in *E. coli* 4W, the gene *glyA* was deleted with CRISPR/Cas9, thus resulting in strain 4WG, which produced 2.01 g/L L-serine with the addition of glycine. Decreasing SHMT activity strongly affects L-serine accumulation was observed in other studies [5–7]. However, in the present experiments, *glyA* deletion resulted in cell growth inhibition and a lower glycerol consumption rate. These results were not completely consistent with Mundhada's study, in which the T1 strain (*E. coli* MG1655 with *tdcG*, *sdaA* and *sdaB* deletion) had a higher glucose consumption rat e and a lower cell growth than the Q1 (strain T1 with *glyA* deletion) [5]. With *glyA* deletion, the cell growth did not significantly change, possibly because of the different carbon source. The low growth rate of strain 4WG limited the its application. To overcome this problem, we used ALE to enhance the strain's tolerance.

ALE or random mutagenesis followed by screening for a non-selectable phenotype is often labor-intensive [32,34]. However, this process can be circumvented by combining ALE with biosensors [23,24]. By using ALE combined with serine-biosensor, we obtained the evolved strain 4WGX, and L-serine production was increased. The maximum $OD_{600}$ of strain 4WGX was 6.87, glycerol was consumed completely at 24 h, faster than the parental strain 4W, with the addition of 2 mM glycine. The L-serine titer was 4.13 g/L at 48 h (Figure 6a), a value 105% higher than that of 4WG and 275% higher than that of 4W. The substrate conversion rate was 41.3%, and the value reported for the strain developed in this study is close to highest yield reported from sugar [5]. Moreover, sequencing results of strain 4WGX revealed a total of eleven single base mutations in the genome of strain 4WGX, including ten non-synonymous mutations and one synonymous mutation (Table S1). Interestingly, most of the mutated genes encoded the membrane protein, such as *brnQ*, encoding the branched chain amino acid transporter BrnQ, *fepB*, *oppB* encoding the ABC transporter, and *fepB*, encoding FepB with a key role in transporting the catecholate siderophore ferric enterobactin from the outer to the inner membrane in Gram-negative

bacteria [35]. *ygbN* encodes a putative transporter. *bamA* encodes the outer membrane protein BamA in *E. coli*, and a recent study has reported that the outer membrane fluidity linked to BamA activity [36,37]. The gene *agp* encodes glucose-1-phosphatase, and *eno* encodes enolase. The function of the other mutated genes (*ybcJ, fepB, dgcT, fliK*) were unclear. We chose to study the genes *agp* and *eno*, which were not involved in the membrane. Reverse mutation of *agp* and *eno* were performed in the genome of 4WG, thus yielding mutant strains. However, we did not observe any significant changes in L-serine production and cell growth in both strains (data not shown). The previous study has showed that a site-specific variant of enolase results in the functional and structural changes [38]. Most of the mutated genes (*brnQ,BrnQ, fepB, oppB, ygbN* and *bamA*) encoded transporters in *E. coli*, and these mutations were likely to alter cellular metabolite to help bacteria cope with the toxic metabolite [37,39], thus potentially explained why the final evolved strain grew better than the parent strain in 50 g/L L-serine. Further studies are needed to explore relationship between the membrane protein and phenotypic.

Biosensors have been widely used to develop high throughput screening methods and optimize pathway expression [40–42]. In our previous study, the serine-biosensor pDser, which was based on NCgl0581 (a transcription factor specifically responsive to L-serine in *C. glutamicum*), had been constructed in *C. glutamicum* for high-throughput screening of L-serine high-yield strains. However, we did not know whether this biosensor was suitable for screening L-serine over-producing *E. coli*, because the heterologous expression of the transcriptional regulator might have significantly interfered with the host gene regulatory networks. Moreover, the sensitivity of the biosensor was determined by the rate of promoter occupation by transcription factors through protein-protein interaction [43]. Therefore, the validity and sensitivity of serine-biosensor pDser were verified, and the results showed that biosensor from *C. glutamicum* was effective in selecting L-serine over-producing *E. coli*. On this basis, we developed a high-throughput screening method. Moreover, in evolution experiments, L-serine was added to the medium, and the effect of the L-serine addition to the biosensor was studied. No significant change in fluorescence intensity was observed with varying amount of L-serine added (data not shown), thus indicating that only the cellular L-serine biosynthesized was monitored by serine-biosensor. This work indicates that the serine-biosensor from *C. glutamicum* is useful in selecting serine over-producing *E. coli*, thus expanding the application of biosensor and enabling expanded strategies for screening high performance strains. Moreover, glycerol is a promising carbon source for the production of L-cysteine. In the future, to further increase L-cysteine production, we will focus on the genes involved in biosynthesis and transport of L-cysteine, and in the uptake of sulfur sources.

**Supplementary Materials:** The following supporting information can be downloaded at: https://www.mdpi.com/article/10.3390/fermentation8070299/s1, Table S1. The eleven single base mutations in the genome of strain 4WGX. References [35,37–39,44–50] are cited in the supplementary materials.

**Author Contributions:** Conceptualization, X.Z. (Xiaomei Zhang); methodology, D.Z.; validation, Z.S. and J.B.; data curation, Y.G.; writing—original draft preparation, X.Z. (Xiaomei Zhang) and D.Z.; writing—review and editing, G.X.; visualization, X.Z. (Xiaojuan Zhang) and H.L.; supervision, J.S.; project administration, Z.X.; funding acquisition, X.Z. (Xiaomei Zhang). All authors have read and agreed to the published version of the manuscript.

**Funding:** This work was financially supported by the National Key Research and Development Program of China (2018YFA0901400). The National Natural Science Foundation of China (32171470).

**Institutional Review Board Statement:** Not applicable.

**Informed Consent Statement:** Not applicable.

**Data Availability Statement:** Not applicable.

**Conflicts of Interest:** The authors declare no conflict of interest.

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
