# Peer review of "Rational Metabolic Engineering Combined with Biosensor-Mediated Adaptive Laboratory Evolution for l-Cysteine Overproduction from Glycerol in Escherichia coli"

_fermentation, doi:10.3390/fermentation8070299_

Round 1

Reviewer 1 Report

The experiments behind the manuscript are sound and well done. The major drawback is the English language, with several style mistakes that can be corrected by a native speaker. 

A minor comment is on the genome sequencing method - some more data on kits used for library prep as well as sequencing, together with  the software parametres used for trimming, QC and variant calling are required.

Author Response

Reviewer #1: Reviewer Comments for:

Q1. The experiments behind the manuscript are sound and well done. The major drawback is the English language, with several style mistakes that can be corrected by a native speaker.

Answer:Thanks for the kind advices of the reviewer. The use of English has been carefully modified based on the advices of a native English speaker.

Q2. A minor comment is on the genome sequencing method - some more data on kits used for library prep as well as sequencing, together with the software parametres used for trimming, QC and variant calling are required.

Answer:Thanks for the kind advices of the reviewer. This information about the genome sequencing method had been provided in revised manuscript. “The whole genome of E. coli 4WGX was sequenced, and comparative genomics analysis was performed with the parent strain 4WG. Genomic DNAs of the strains were extracted using Molpure Bacterial DNA Kit (Yeasen. Shanghai, China). Library construction and genome sequencing were performed by Genewiz (Suzhou, China) by using Illumina Hiseq2500 sequencing platform. Quality assurance of the output was analyzed by using FastQC software (v.0.10.1) and NGSQC Toolkit software (v.2.3.3). BWA alignment software (v.0.7.17) and SAM tools software (v.1.9) were used for alignment and variant calling, respectively. Variations were annotated by using the SnpEff software (v.4.3i).”

Please see P8L12- P8L20 in the revised section “Materials and Methods”.

Reviewer 2 Report

The manuscript of Zhang et al. describes metabolic engineering efforts toward production of cysteine. Previously, the authors developed a production platform for serine based on C. glutamicum, including a sensor system. The research group also developed a sensor system for testing overproduction of serine in that strain. In the presented work, the authors transferred the serine production platform to E. coli thereby aiming biosynthesis of cysteine starting from glycerol. They disabled two major serine degradation pathways, and performed an adaptive laboratory evolution of resulting strain toward a higher yield of serine. The sensor system was utilized for final selection of the candidate strains. The authors achieved an overproduction of serine up to several grams per litre culture. In the final step, the authors appended a serine to cysteine metabolic conversion, and achieved an overproduction of serine up to several hundreds of microgram cystein per litre culture.

The key challenge of the presented work was an evolutionary adaptation of strains towards an accumulation of intracellular serine. Serine is a metabolite with very well known toxic effects on bacteria. By using serial passages of the strain from exponentially growing phase into fresh media, the authors evolved an overproducing strain. The genetic analysis of resulting genome indicated 11 key mutations.

I have almost no reservations regarding the presented manuscript. However, the design of the study seemed overall dubious to me. Main efforts were put into selecting a serine overproducing strain, while it was not the intended target product of the study. Serine should not be required at high intracellular concentration, since this substance is only an intermediate metabolite for the cycteine production. Thus, most efforts in generating a highly producing serine strain seems like an overkill. I hope, in the future works, the authors will change their focus from intermediates to the final product, which would create a far more adequate design of the study.

Nonetheless, the presented manuscript is still quite interesting to read. The information about key mutations and the potential of serine and cystein overproduction in E. coli represent a scientific interest. I am in favor of the publication of the manuscript with a minor revision:

1.  Please, provide details on the progress of ALE experiment: how many passages were made, how long was the experiment conducted, how many repeats, how was the progress monitored and whether or not the authors noted any interesting observations along the process. For example, was evolutionary adaptation achieved gradually or stepwise?

2. Please, provide some more discussion of the 11 mutated genes. Is it possible to speculate what could be the roles of these mutations? Are they needed to enhance production/reduce degradation of serine, or their role to help bacteria cope with an otherwise toxic metabolite?

3. Please, provide some specific details for the sensor system. It is a little unclear how it works and how is fluorescent response was generated. 

I thank the authors for an interesting read and with them best! 

Author Response

Reviewer #2: Reviewer Comments for:

The manuscript of Zhang et al. describes metabolic engineering efforts toward production of cysteine. Previously, the authors developed a production platform for serine based on C. glutamicum, including a sensor system. The research group also developed a sensor system for testing overproduction of serine in that strain. In the presented work, the authors transferred the serine production platform to E. coli thereby aiming biosynthesis of cysteine starting from glycerol. They disabled two major serine degradation pathways, and performed an adaptive laboratory evolution of resulting strain toward a higher yield of serine. The sensor system was utilized for final selection of the candidate strains. The authors achieved an overproduction of serine up to several grams per litre culture. In the final step, the authors appended a serine to cysteine metabolic conversion, and achieved an overproduction of serine up to several hundreds of microgram cystein per litre culture. 

The key challenge of the presented work was an evolutionary adaptation of strains towards an accumulation of intracellular serine. Serine is a metabolite with very well known toxic effects on bacteria. By using serial passages of the strain from exponentially growing phase into fresh media, the authors evolved an overproducing strain. The genetic analysis of resulting genome indicated 11 key mutations. 

I have almost no reservations regarding the presented manuscript. However, the design of the study seemed overall dubious to me. Main efforts were put into selecting a serine overproducing strain, while it was not the intended target product of the study. Serine should not be required at high intracellular concentration, since this substance is only an intermediate metabolite for the cycteine production. Thus, most efforts in generating a highly producing serine strain seems like an overkill. I hope, in the future works, the authors will change their focus from intermediates to the final product, which would create a far more adequate design of the study. 

Nonetheless, the presented manuscript is still quite interesting to read. The information about key mutations and the potential of serine and cystein overproduction in E. coli represent a scientific interest. I am in favor of the publication of the manuscript with a minor revision:

Thanks for the kind advices of the reviewer. In L-cysteine biosynthetic pathway, L-serine is the precursor for the L-cysteine production,to construct overproducing L-serine strain was the key to increase target product L-cysteine titer. Then the efforts were put into selecting a serine overproducing strain in this study, and L-cysteine producing strain had been constructed by using glycerol as the carbon source. In the future works, according to the kind advices of the reviewer, to increase the final product L-cysteine production further, we will create a far more adequate design to focus L-cysteine,such as L-cysteine biosynthetic pathway and transport pathway.

Q1. Please, provide details on the progress of ALE experiment: how many passages were made, how long was the experiment conducted, how many repeats, how was the progress monitored and whether or not the authors noted any interesting observations along the process. For example, was evolutionary adaptation achieved gradually or stepwise?

Answer:Thanks for the kind advices of the reviewer, the details of ALE experiment have been provided in the section “Materials and Methods” of the revised manuscript. “For the strain evolution, E. coli 4WG harboring pDser was cultured in LB medium, transferring 10% (v/v) inoculum into fresh AM1 medium with 6 g/L L-serine for 24 h,and repeated this step 10 times (recorded as 10 generations), actually, the evolved strain was about 600 generations. Then, increase L-serine concentration to 12, 25 and 50 g/L respectively, repeated this process. Successive rounds of ALE were carried out with the L-serine increased stepwise (6, 12, 25 and 50 g/L). According to the generation time (GT) of E. coli, the evolved strain was about 600, 1200, 1800, 2400 generations, respectively. At the end of the experiment, an appropriate amount solution of the evolved strains of 600, 1200, 1800, 2400 generations were diluted to 10-5-10-6 times. Then 100 μL of diluted bacterial solution was spread on kanamycin plates and cultured overnight at 37℃, and single colonies were transferred to 96-well plates. After 24 h of fermentation, the fluorescence intensity of the strain in each well was measured, the most efficient strain was selected by using FACS according to our previous study.” Please see P7L19- P8L10 in the revised section “Materials and Methods”.

Q2. Please, provide some more discussion of the 11 mutated genes. Is it possible to speculate what could be the roles of these mutations? Are they needed to enhance production/reduce degradation of serine, or their role to help bacteria cope with an otherwise toxic metabolite?

Answer:Thanks for the kind advices of the reviewer. Some more discussion of the 11 mutated genes had been provided in the section“Discussion”. “Interestingly, most of the mutation gene linked to the membrane protein, such as brnQ, encoding the branched chain amino acid transporter BrnQ, fepB, oppB encoding the ABC transporter, and fepB,encoding FepB with a key role in transporting the catecholate siderophore ferric enterobactin from the outer to the inner membrane in Gram-negative bacteria [35]. ygbN encoded the putative transporter, bamA encoded the outer membrane protein BamA in E. coli, and the recent study reported that the outer membrane fluidity linked to BamA activity. The gene agp encoded glucose-1-phosphatase and eno encoded enolase. The other mutation gene’s (ybcJ, fepB, dgcT, fliK) function were unclear. We chose to study the two genes agp and eno not involved in the membrane firstly, the reverse mutation of agp and eno were performed in the genome of 4WG, thus resulting in the mutant strains. However, we did not observe any significant change regarding to L-serine production and the cell growth in both strains (data not shown). Although the previous study showed that site-specific variant of enolase resulted in the functional and structural changes. Most of the mutation gene(brnQ,BrnQ, fepB, oppB, ygbN and bamA)were transporter in E.coli, and these mutations most likely result in the change of cellular metabolite, then to help bacteria cope with an otherwise toxic metabolite. This might be the reason why the final evolved strain grew better than the parent strain in 50 g/L L-serine.”

Please see P16L8- P17L4 in the revised section “Discussion”.

Q3. Please, provide some specific details for the sensor system. It is a little unclear how it works and how is fluorescent response was generated.

Answer:Thanks for the kind advices of the reviewer. To make it clearer and easy to understand, we have provided some specific details for the sensor system in the section “Introduction”. “High-throughput screening methods with biosensors had been used to screen mutant strains, such as those that overproduce amino acids and organic acids, and the biosensor devices, including riboswitches, enzymes, and transcription factors can transform information about a specific metabolite into a graded fluorescence output. In our previous study, we developed a genetic metabolite biosensor capable of detecting L-serine in single C. glutamicum cells, this biosensor is based on the transcriptional regulator NCgl0581 of C. glutamicum, which activates expression of the NCgl0580 promoter to drive transcription of enhanced yellow fluorescent protein. However, no study had reported screening of L-serine overproducing strains by using a biosensor in E. coli.”

Please see PL18- P5L5 in the revised section “Introduction”.

Round 2

Reviewer 2 Report

I thank the authors for addressing my comments. Best luck! 

Author Response

Thank you very much for your support and help.